# Self-Training: A Survey

## Abstract

Semi-supervised algorithms aim to learn prediction functions from a small set of labeled training set and a large set of unlabeled observations. Because these approaches are relevant in many applications, they have received a lot of interest in both academia and industry. Among the existing techniques, self-training methods have undoubtedly attracted greater attention in recent years. These models are designed to find the decision boundary on low density regions without making additional assumptions about the data distribution, and use the unsigned output score of a learned classifier, or its margin, as an indicator of confidence. The working principle of self-training algorithms is to learn a classifier iteratively by assigning pseudo-labels to the set of unlabeled training samples with a margin greater than a certain threshold. The pseudo-labeled examples are then used to enrich the labeled training data and to train a new classifier in conjunction with the labeled training set. In this paper, we present self-training methods for binary and multi-class classification as well as their variants and two related approaches, namely consistency-based approaches and transductive learning. We also provide brief descriptions of self-supervised learning and reinforced self-training, two distinct approaches despite their similar names. Finally, we present the most popular applications where self-training is employed. For pseudo-labeling, fixed thresholds usually lead to subpar results, highlighting the significance of dynamic thresholding for best results. Moreover, improving pseudo-label noise enhances generalization and class differentiation. The performance is also impacted by augmenting initial labeled training samples. These findings highlight the complex interplay in self-training efficacy between threshold selection, noise control, and labeled training size. They emphasize the need for meticulous parameter tuning and data preprocessing to fully exploit semi-supervised learning's potential and pave the way for future research in refining methodologies and expanding applicability across domains. To the best of our knowledge, this is the first thorough and complete survey on self-training.

## 1 Introduction

Semi-supervised learning has risen to prominence within the machine learning domain, tackling the core challenge of making inference from both the structure of the unlabeled data and the label information from existing labeled training sets (Altun et al., 2005). The framework is particularly useful in scenarios where there are limited labeled examples but an abundance of unlabeled data available for training. This is highly relevant in a range of applications, such as computer vision, natural language processing and speech recognition, where the acquisition of labeled data can be a costly endeavor (Yu et al., 2022; Cheng et al., 2023; Gheini et al., 2023; Qu et al., 2023).

### 1.1 Central hypothesis

In general, it remains unclear how unlabeled data can be used in training and what value it can bring. The basic assumption in semi-supervised learning, called *smoothness*, stipulates that two examples in a high density region should have identical class labels (Chapelle et al., 2010; Amini & Usunier, 2015). This means that if two points are part of the same group or cluster, their class labels will most likely be the same. If they are separated by a low density zone, on the other hand, their desired labels should be different. Hence, if the examples of the same class form a partition, unlabeled training data might aid in determining the partition boundary more efficiently than if just labeled training examples were utilized.

## 1.2 Three main semi-supervised learning families

There are three main families of semi-supervised methods, each with its own adaptation of the smoothness hypothesis. These adaptations are usually referred to as assumptions, albeit loosely, since they rather represent different paradigms for implementing semi-supervised learning.

Data clustering uses a mixture model and assigns class labels to groups using the labeled data they include; and it constitutes the working principle of generative approaches (Nigam et al., 2006; Kingma et al., 2014). The cluster assumption, which underpins these approaches, asserts that if two examples are in the same group, they are likely to be in the same class (Figure 1 (a)). This hypothesis may be explained as follows: if a group is formed by a large number of instances, it is rare that they belong to different classes. This does not imply that a class is constituted by a single group of examples, but rather that two examples from distinct classes are unlikely to be found in the same cluster (Abu-Mostafa, 1995).

If we consider the partitions of instances to be high density areas, a form of the cluster assumption known as low density separation entails determining the decision boundary over low density regions (Figure 1 (b)), and it constitutes the basis of discriminant techniques. The main difference between generative and discriminant techniques is that discriminant approaches find directly the prediction function without making any assumption on how the data are generated (Amini & Gallinari, 2002; Grandvalet & Bengio, 2004; Oliver et al., 2018).

Density estimation is often based on a notion of distance, which may become meaningless for high dimensional spaces. A third hypothesis, known as the manifold assumption, stipulates that in high-dimensional spaces, instances reside on low-dimensional topological spaces that are locally Euclidean (Figure 1 (c)), which is supported by a variety of semi-supervised models called graphical approaches (Belkin & Niyogi, 2004; Chong et al., 2020).

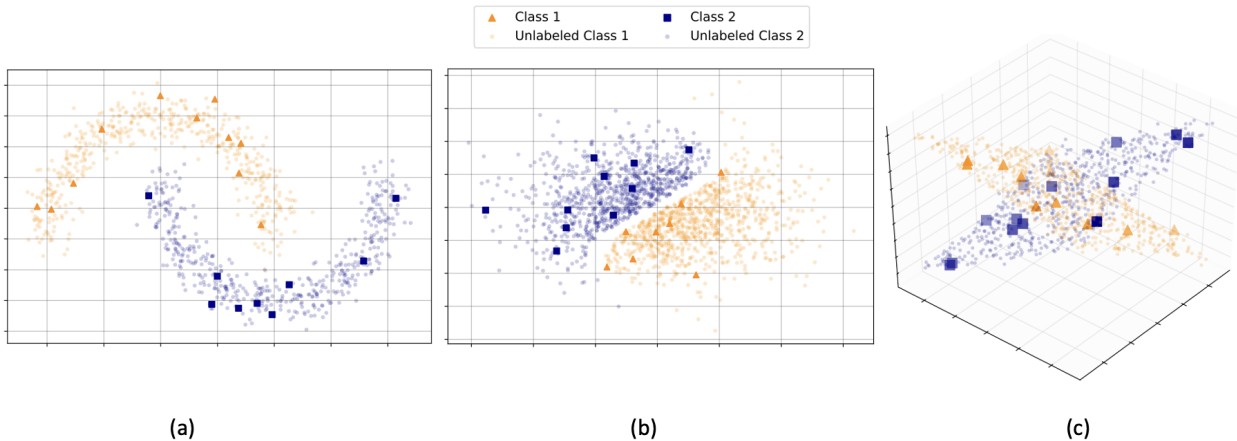

Figure 1: Illustration of three main hypotheses made in semi-supervised learning: (a) cluster assumption, (b) low-density separation and (c) manifold assumption.

## 1.3 Compatibility

Although semi-supervised algorithms have been successfully applied in many situations, there have been cases where unlabeled data have been shown to have no effect on the performance of a learning task (Singh et al., 2008). Several attempts have been made in recent years to investigate the value of unlabeled data in the training process (Castelli & Cover, 1995; Li & Zhou, 2011), and the capacity of semi-supervised learning approaches to generalize (Rigollet, 2007; Maximov et al., 2018). The bulk of these studies are founded on the notion of *compatibility* defined by Balcan & Blum (2006), and they strive to exhibit the connection between the marginal data distribution and the target function to be learned. According to these findings, unlabeled data will be beneficial for training only if such a relationship exists.

In generative approaches, the marginal distribution is viewed as a mixture of class conditional distributions, and when compared to the supervised case, semi-supervised learning has been shown to achieve lower finite-sample error bounds

in some general cases, or a faster rate of error convergence in others (Castelli & Cover, 1995; Rigollet, 2007; Maximov et al., 2018; Singh et al., 2008). In this line, Ben-David et al. (2008) showed that accessing the marginal distribution on unlabeled training data would not provide sample size guarantees superior to those obtained by supervised learning unless very strong assumptions about conditional distribution on class labels are made.

For graph-based approaches, Niyogi (2013); Altun (2005) provided a context in which such algorithms may be studied and perhaps justified; the key finding of the study is that unlabeled data can help learning in some situations by explicitly defining the structure of the data through a manifold.

Finally, discriminant approaches mostly embed a margin maximization method that searches the decision boundary in low-density regions by pushing it from the unlabeled data (Joachims, 1999). In this survey we focus on self-training algorithms that follow this principle by assigning pseudo-labels to high-confidence unlabeled training examples and include these pseudo-labeled samples in the learning process. While various surveys have explored semi-supervised learning in recent years (Van Engelen & Hoos, 2020; Yang et al., 2023), none have specifically emphasized self-training, which has emerged as the predominant approach in the field, widely applied across various applications.

### 1.4 Paper structure

The reminder of this paper is organized as follows.

In Section 2, we go over the self-training method in detail. First, we present the framework and notations used throughout the paper in Section 2.1, then we describe the general self-training algorithm in Section 2.2, also introduced in Algorithm 1. Then, we describe pseudo-labeling methods and its variants in Section 2.3, and we discuss the self-training with two classifiers in Section 2.4. Those methods are summed up in Table 1. Finally, we provide some insights into current theoretical studies in Section 2.6.

Other related approaches are described in Section 3. First, we detail the transductive learning context in Section 3.1, and the consistency-based approaches in Section 3.2. Going beyond traditional semi-supervised learning, we investigate the extension of self-training in domain adaptation in Section 2.5, delve into self-supervised learning in Section 3.3, and explore reinforced self-training in Section 3.4.

Section 4 reviews application of self-training methods in different domains, such as natural language processing in Section 4.1, computer vision in Section 4.2 and more generally in knowledge-driven applications in Section 4.3, with speech recognition, anomaly detection and genomics and proteomics.

The views and future prospects are discussed in Section 5.

## 2 Self-Training

Within this section, we present the fundamental aspects of the self-training approach. Initially, we introduce the framework and notation, followed by a comprehensive exploration of the core concept behind the self-training algorithm, which is further delineated in Algorithm 1. In Section 2.3 and Section 2.4, we present significant contributions directly linked to the standard algorithm. We organize these contributions effectively in Table 1. To conclude, we delve into the theoretical aspects in Section 2.6.

### 2.1 Semi-supervised framework and notations

We consider classification problems where the input and the output spaces are respectively $\mathcal{X} \subseteq \mathbb{R}^d$ and $\mathcal{Y} = \{-1, +1\}$ or $\mathcal{Y} = \{1, \ldots, K\}$. We further suppose available a set of labeled training examples $S = (\mathbf{x}_i, y_i)_{1 \leqslant i \leqslant m} \in (\mathcal{X} \times \mathcal{Y})^m$ generated from a joint probability distribution $\mathbb{P}(\mathbf{x}, y)$ (denoted as $\mathcal{D}$) and a set of unlabeled training examples $X_{\mathcal{U}} = (\mathbf{x}_i)_{m+1 \leqslant i \leqslant m+u} \in \mathcal{X}^u$ supposed to be drawn from the marginal distribution $\mathbb{P}(\mathbf{x})$.

The classic case corresponds to when $m \ll u$, and the issue is thrown into the unsupervised learning framework if $S$ is empty. The opposite extreme scenario is when $X_{\mathcal{U}}$ is empty and the problem is reduced to supervised learning. Given a hypothesis set of functions $\mathcal{H}$ mapping $\mathcal{X}$ to $\mathcal{Y}$, the learner receives a labeled set $S$ and an unlabeled set $X_{\mathcal{U}}$ and outputs a hypothesis $h \in \mathcal{H}$ which is assumed to have a generalization error $R(h) = \mathbb{E}_{(\mathbf{x}, y) \sim \mathcal{D}}[\mathbb{1}_{h(\mathbf{x}) \neq y}]$ smaller

than if just $S$ was used to find the prediction function, where by $\mathbb{1}_\pi$ we denote the indicator function equal to 1 if the predicate $\pi$ is true and 0 otherwise.

In practice, classifiers are defined based on a scoring function $f$ from a class of functions $\mathcal{F} = \{f : \mathcal{X} \times \mathcal{Y} \to \mathbb{R}\}$, and for an example $\mathbf{x}$ the corresponding classification function $h$ outputs the class for which the score of $f$ is the highest:

$$h(\mathbf{x}) = \mathrm{argmax}_{y \in \mathcal{Y}} f(\mathbf{x}, y).$$

We define the margin $\rho_f(\mathbf{x}, y)$ of a function $f$ for an example $\mathbf{x} \in \mathcal{X}$ and a class $y \in \mathcal{Y}$ as

$$\rho_f(\mathbf{x}, y) = f(\mathbf{x}, y) - \max_{y' \neq y} f(\mathbf{x}, y').$$

In the binary case, $\mathcal{Y} = \{-1, +1\}$, we define the unsigned margin of a classification function $f \in \mathcal{F}$ over an example $\mathbf{x} \in \mathcal{X}$ (d'Alché Buc et al., 2001; Amini et al., 2008) as

$$m_f(\mathbf{x}) = |\rho_f(\mathbf{x}, +1)|.$$

In the multi-class classification case, $\mathcal{Y} = \{1, \dots, K\}$, the unsigned margin (d'Alché Buc et al., 2001; Feofanov et al., 2019) is defined as

$$m_f(\mathbf{x}) = \sum_{y \in \mathcal{Y}} f(\mathbf{x}, y) \rho_f(\mathbf{x}, y).$$

The maximization of the unsigned margin tends to find a decision boundary that passes through low density regions and hence follows the low density separation assumption.

## 2.2 Self-training: the idea

Self-training, also known as decision-directed or self-taught learning machine, is one of the earliest approach in semi-supervised learning (Scudder, 1965; Fralick, 1967) that has risen in popularity in recent years.

To determine the decision boundary on low density regions, the idea behind self-training algorithms is to consider a pseudo-labeling strategy for assigning pseudo-labels to the examples of $X_\mathcal{U}$. This strategy can be characterized by a function, called *pseudo-labeler*:

$$\Phi_\ell : \mathcal{X} \times \mathcal{F} \to \mathcal{X} \times \mathcal{Y}.$$

We denote $\tilde{y}$ the pseudo-label of an unlabeled $\mathbf{x} \in X_\mathcal{U}$ for a score function $f \in \mathcal{F}$ assigned by $\Phi_\ell$ and $X_{\overline{\mathcal{U}}}$ the set of pseudo-labeled examples.

The self-learning strategy is an iterative wrapper algorithm that starts by learning a supervised classifier on the labeled training set $S$. Then, at each iteration, the current classifier selects a part of the unlabeled data, $X_{\overline{\mathcal{U}}}$, and assigns pseudo-labels to them using the classifier's predictions.

These pseudo-labeled unlabeled examples are removed from $X_\mathcal{U}$ and a new supervised classifier is trained over $S \cup X_{\overline{\mathcal{U}}}$, by considering these pseudo-labeled unlabeled data as additional labeled examples. To do so, the classifier $h \in \mathcal{H}$ that is learned at the current iteration is the one that minimizes a regularized empirical loss over $S$ and $X_{\overline{\mathcal{U}}}$:

$$\frac{1}{m} \sum_{(\mathbf{x}, y) \in S} \ell(h(\mathbf{x}), y) + \frac{\gamma}{|X_{\overline{\mathcal{U}}}|} \sum_{(\mathbf{x}, \tilde{y}) \in X_{\overline{\mathcal{U}}}} \ell(h(\mathbf{x}), \tilde{y}) + \lambda \|h\|^2$$

where $\ell : \mathcal{Y} \times \mathcal{Y} \to [0, 1]$ is an instantaneous loss most often chosen to be the cross-entropy loss, $\gamma$ is a hyperparameter for controlling the impact of pseudo-labeled data in learning, and $\lambda$ is the regularization hyperparameter. This process of pseudo-labeling and learning a new classifier continues until the unlabeled set $X_\mathcal{U}$ is empty or there is no more unlabeled data to pseudo-label. The pseudo-code of the self-training algorithm is shown in Algorithm 1.

## 2.3 Pseudo-labeling strategies

At each iteration, the self-training selects just a portion of unlabeled data for pseudo-labeling, otherwise, all unlabeled examples would be pseudo-labeled after the first iteration, which would actually result in a classifier with performance

---

**Algorithm 1. Self-Training**

> **Input :** $S = (\mathbf{x}_i, y_i)_{1 \leqslant i \leqslant m}$, $X_{\mathcal{U}} = (\mathbf{x}_i)_{m+1 \leqslant i \leqslant m+u}$.
> $k \leftarrow 0$, $X_{\bar{\mathcal{U}}} \leftarrow \emptyset$.
> **repeat**
>   Train $f^{(k)}$ on $S \cup X_{\bar{\mathcal{U}}}$
>   $\Pi_k \leftarrow \{\Phi_\ell(\mathbf{x}, f^{(k)}), \mathbf{x} \in X_{\mathcal{U}}\}$   $\triangleright$ Pseudo-labeling
>   $X_{\bar{\mathcal{U}}} \leftarrow X_{\bar{\mathcal{U}}} \cup \Pi_k$
>   $X_{\mathcal{U}} \leftarrow X_{\mathcal{U}} \setminus \{\mathbf{x} \mid (\mathbf{x}, \tilde{y}) \in \Pi_k\}$
>   $k \leftarrow k + 1$
> **until** $X_{\mathcal{U}} = \emptyset \vee \Pi_k = \emptyset$
> **Output :** $f^{(k)}, X_{\mathcal{U}}, X_{\bar{\mathcal{U}}}$

---

identical to the initial classifier (Chapelle et al., 2010). Thus, the implementation of self-training arises the following question: how to determine the subset of examples to pseudo-label?

A classical assumption, that stems from the low density separation hypothesis, is to suppose that the classifier learned at each step makes the majority of its mistakes on observations close to the decision boundary. In the case of binary classification, preliminary research suggested to assign pseudo-labels only to unlabeled observations for which the current classifier is the most confident (Tür et al., 2005). Hence, considering thresholds $\theta^-$ and $\theta^+$ defined for respectively the negative and the positive classes, the pseudo-labeler assigns a pseudo-label $\tilde{y}$ to an instance $\mathbf{x} \in X_{\mathcal{U}}$ such that:

$$\tilde{y} = \begin{cases} +1, & \text{if } f(\mathbf{x}, +1) \geqslant \theta^+, \\ -1, & \text{if } f(\mathbf{x}, -1) \leqslant \theta^-, \end{cases} \tag{1}$$

and $\Phi_\ell(\mathbf{x}, f) = (\mathbf{x}, \tilde{y})$. An unlabeled example $\mathbf{x}$ that does not satisfy the conditions equation 1 is not pseudo-labeled; i.e. $\Phi_\ell(\mathbf{x}, f) = \emptyset$.

Intuitively, thresholds should be set to high absolute values as pseudo-labeling examples with low confidence would increase chances of assigning wrong labels. However, thresholds of very high value imply excessive trust in the confidence measure underlying the model, which, in reality, can be biased due to the small labeled sample size. Using several iterations makes also the situation more intricate as at every iteration the optimal threshold might be different.

One way to select the thresholds is to set them equal to the average of respectively positive and negative predictions (Tür et al., 2005). In this line, and in the context of multi-class classification (Lee, 2013) used Neural Networks as the supervised classifier and chose the most confident class to infer pseudo-labels for unlabeled data using the current model' outputs. The pseudo-labeled examples were then added to the labeled training set and treated similarly as labeled examples.

Zou et al. (2018) adapted the idea of Tür et al. (2005) for multi-class classification by not choosing thresholds but rather fixing a proportion $p$ of the most confident unlabeled data to be pseudo-labeled and then increasing this proportion at each iteration of the algorithm until $p = 0.5$ was reached. Following this idea, Cascante-Bonilla et al. (2021) revisited the concept of pseudo-labeling by discussing the iterative process of assigning pseudo-labels to unlabeled data and emphasized the resilience of pseudo-labeling to out-of-distribution samples.

Zhang et al. (2021) also proposed an adaptation of curriculum learning to pseudo-labeling, which entails in learning a model using easy-to-learn observations before moving on to more complex ones. The principle is that at the step $k$ of the algorithm, the pseudo-labeler selects unlabeled examples having predictions that are in the $(1 - \alpha_k)^{th}$ percentile of the distribution of the maximum probability predictions assumed to follow a Pareto distribution, and where $\alpha_k \in [0, 1]$ is an hyperparameter that varies from 0 to 1 in increments of $0.2$.

Considering the distribution of predictions over unlabeled data, and the majority vote classifiers, such as Random Forest or Adaboost (Schapire et al., 1997), it is possible to automatically choose a threshold for pseudo-labeling. Formally, the learning of a majority vote classifier with partially labeled data can be defined as follows.

After observing the training set $S \cup X_{\overline{\mathcal{U}}}$, the task of the learner is to choose a posterior distribution $Q$ over a set of hypothesis $\mathcal{H}$ such that the $Q$-*weighted majority vote* classifier $B_Q$ defined by:

$$\forall \mathbf{x} \in \mathcal{X}, B_Q(\mathbf{x}) = \operatorname{argmax}_{y \in \mathcal{Y}} \mathbb{E}_{h \sim Q}\left[\mathbb{1}_{h(\mathbf{x})=y}\right],$$

will have the smallest possible risk on examples of $X_{\mathcal{U}}$. The associated *Gibbs* classifier, $G_Q$, is defined as the random choice of a classifier $h$ according to the distribution $Q$, and its error over an unlabeled set $X_{\mathcal{U}}$ is given by:

$$\hat{R}_u(G_Q) = \frac{1}{u} \sum_{\mathbf{x}' \in X_{\mathcal{U}}} \mathbb{E}_{h \sim Q}[\mathbb{1}_{h(\mathbf{x}') \neq y'}],$$

where, for every unlabeled example $\mathbf{x}' \in X_{\mathcal{U}}$ we refer to $y'$ as its true unknown class label. For binary and multi-class classification respectively, Amini et al. (2008) and Feofanov et al. (2019) showed that a tight upper-bound on the Gibbs classifier's risk that holds with high probability over the random choice of $S$ and $X_{\mathcal{U}}$, guarantees a tight bound on the error of the Bayes classifier over the unlabeled training set:

$$\hat{R}_u(B_Q) = \frac{1}{u} \sum_{\mathbf{x}' \in X_{\mathcal{U}}} \mathbb{1}_{B_Q(\mathbf{x}') \neq y'}.$$

This bound is mainly based on the distribution of predictions over unlabeled data and the derivations can be extended to bound the risk of voted majority classifiers having margins greater than a threshold $\theta$, $\hat{R}_{u \wedge \theta}(B_Q)$, defined as:

$$\hat{R}_{u \wedge \theta}(B_Q) = \frac{1}{u} \sum_{\mathbf{x}' \in X_{\mathcal{U}}} \mathbb{1}_{B_Q(\mathbf{x}') \neq y' \wedge m_{B_Q}(\mathbf{x}') > \theta},$$

with a slight abuse of notation for $m_{B_Q}$. One of the practical aspects that arises from this result is the possibility to specify a threshold $\theta$ which minimizes an upper-bound of the conditional risk of a voted majority classifier $B_Q$ over the unlabeled training set, $X_{\mathcal{U}}$, defined as:

$$\hat{R}_{u|\theta}(B_Q) = \frac{\hat{R}_{u \wedge \theta}(B_Q)}{\frac{1}{u} \sum_{\mathbf{x} \in X_{\mathcal{U}}} \mathbb{1}_{m_{B_Q}(\mathbf{x}) \geqslant \theta}},$$

where the denominator is the proportion of the unlabeled examples with the confidence higher than the threshold $\theta$, and the numerator is the joint Bayes risk on $X_{\mathcal{U}}$. Thus, the criterion can be interpreted as a trade-off between the number of examples going to be pseudo-labeled and the error they induce. Furthermore, these bounds are shown to be tight in the case where the majority vote classifier makes its error mostly on low margin regions. Feofanov et al. (2019) demonstrated that this technique outperforms conventional fixed-threshold pseudo-labeling strategies on different multi-class classification problems.

Chen et al. (2022) highlighted two major issues with self-training: the snowball effects of cascading pseudo-labeling mistakes and random sampling of tiny samples (called data bias). The authors suggest two-phase solutions to address these problems for image classification using deep learning. First, they proposed a classification head to separate the creation and use of pseudo labels in order to reduce training errors. An additional head is utilized to receive the pseudo-labels and carry out training on unlabeled data while the default head is used for classification and pseudo-labeling.

## 2.4 Self-training with two classifiers

In the wake of works utilizing only a single classifier in self-training algorithms, new studies have been proposed with the use of two classifiers, where each model learns on the output of the other (Xie et al., 2020b; Chen et al., 2021; Karamanolakis et al., 2021). Most of these techniques are based on the idea of consensus in predictions between two classifiers and were inspired by the seminal work of Blum & Mitchell (1998) who proposed the co-training algorithm.

In co-training, examples are defined by two modalities that are comparable but not entirely correlated. Each view of an example is expected to contain complementary information about the data and if there are enough labeled training data, each of them is supposed to be sufficient for learning. The main principle is to learn a classifier on each view,

| | Base classifier | | Classification | | Threshold | | Noise |
|---|---|---|---|---|---|---|---|
| | Single | Double | Binary | Multi-class | Fixed | Optimized | Account |
| Scudder [1965] | ✓ | − | ✓ | − | ✓ | − | − |
| Joachims. [1999] | ✓ | − | ✓ | − | ✓ | − | − |
| Amini et al. [2008] | ✓ | − | ✓ | − | − | ✓ | − |
| Hadjadj et al. [2023] | ✓ | − | ✓ | − | − | ✓ | ✓ |
| Tur et al. [2005] | ✓ | − | − | ✓ | ✓ | − | − |
| Xie et al. [2020a] | ✓ | − | − | ✓ | ✓ | − | − |
| Cascante et al. [2021] | ✓ | − | − | ✓ | ✓ | − | − |
| Chen et al. [2022] | ✓ | − | − | ✓ | ✓ | − | ✓ |
| Feofanov et al. [2019] | ✓ | − | − | ✓ | − | ✓ | − |
| Zhang et al. [2021] | ✓ | − | − | ✓ | − | ✓ | − |
| Blum et al. [1998] | − | ✓ | ✓ | − | ✓ | − | ✓ |
| Yaslan et al. [2010] | − | ✓ | − | ✓ | ✓ | − | − |
| Tarvainen and Valpola [2017] | − | ✓ | − | ✓ | ✓ | − | − |
| Xie et al. [2020b] | − | ✓ | − | ✓ | ✓ | − | − |
| Karamanolakis et al. [2021] | − | ✓ | − | ✓ | ✓ | − | − |
| Chen et al. [2021] | − | ✓ | − | ✓ | ✓ | − | − |
| Ghiasi et al. [2021] | − | ✓ | − | ✓ | ✓ | − | − |
| Du et al. [2022] | − | ✓ | − | ✓ | ✓ | − | − |

Table 1: A summary of principal self-training algorithms, based on pseudo-labeling with one or two classifiers, introduced in Section 2.3 and 2.4.

taking initially the available labeled examples as the training set. Then, one of the classifiers assigns pseudo-labels to unlabeled data, and the other one uses them to retrain the model by including them into its training set. At each iteration, the classifiers alternately switch their roles, thereby co-training each other. As for self-training algorithms with a single classifier, this procedure continues until there are no more unlabeled instances to be pseudo-labeled. In practice, several studies artificially generated the two modalities for classification problems where examples are *mono-viewed* and described by a vector representation. These approaches create the two modalities out of one by selecting at random the set of features that should correspond to each modality from the initial set of features; and their efficiency was empirically proved on various applications (Yaslan & Cataltepe, 2010). Co-training can thus be thought of as a form of self-training: rather than training a classifier on its own previous predictions, each classifier is trained on the predictions of another classifier that was trained on the predictions of the former. Without splitting the input feature set, Chen et al. (2021) proposed Cross Pseudo Supervision for semantic segmentation in images. This method employs two neural networks as supervised classifiers having the same images as inputs. Each neural-network is learned at every mini-batch by considering the pseudo-labels generated by the other network for unlabeled instances as ground-truths. In multi-task learning, Ghiasi et al. (2021) proposed to independently train specialized teachers using labeled datasets. These teachers then label an unlabeled dataset, creating a multitask pseudo-labeled dataset. Subsequently, a student model is trained using the pseudo-labeled dataset, employing multi-task learning to learn from various datasets and tasks simultaneously. Finally, the feature representations of the student model are evaluated across six vision tasks, including image recognition, to assess its performance and generalization capabilities.

The learnability of co-training was studied under the PAC framework (Valiant, 1984), which also accounts for noise in the class labels of unlabeled examples caused by pseudo-labeling. The injection of noisy labels in the pseudo-labeling step is in fact inherent to any self-training algorithm. Taking into account noisy labels in training a model was first considered in supervised learning, following the paradigm of learning with an imperfect supervisor in which training data contains an unknown portion of imperfect labels (Natarajan et al., 2013; Han et al., 2018). Most of these studies tackle this problem from an algorithmic point of view, employing regularization (Miyato et al., 2019) or estimating mislabeling errors by modeling the transition probability between noisy and true labels (Patrini et al., 2017).

Table 1 summarizes the main self-training approaches presented so far by emphasizing their key aspects.

## 2.5   Self-training under Domain Shift

Recently, self-training has expanded its scope beyond semi-supervised learning and has found extensive application to the learning problems where available data is subject to a distribution shift. In unsupervised domain adaptation, where the objective is to transfer knowledge from a labeled source domain to an unlabeled target one, self-training

become a popular alternative to discrepancy minimization methods (Ganin et al., 2016). In this case, self-training aims to progressively correct the domain shift by including more and more pseudo-labeled target examples to the source training set. This is particularly relevant for gradual domain adaptation, where unlabeled instances from intermediate domains are available (Shi & Liu, 2024).

When intermediate domains are not given, it is important to ensure that pseudo-labeled target examples are reliable and are not biased towards the source data. While Inoue et al. (2018) and Zou et al. (2018) approached this issue by carefully choosing a pseudo-labeling policy, Saito et al. (2017) learn a representation via a tri-training scheme, in which the student is trained on target data pseudo-labeled by agreement of two teachers. Liu et al. (2021) alternate between two gradient steps: (1) to train a source classification head that generates pseudo-labels, (2) to train a target classification head using pseudo-labeled data under the constraint that it predicts well on source data.

As the discrepancy between the source and the target can be large, the prediction confidence may exhibit a strong bias failing to distinguish between correct and wrong pseudo-labels. Therefore, several works focus specifically on model calibration and uncertainty estimation including the negative entropy regularization (Zou et al., 2019), the Monte-Carlo dropout (Mukherjee & Awadallah, 2020), and prediction agreement of diversified linear heads (Odonnat et al., 2024).

## 2.6 Theoretical studies

Several studies have recently looked into the theoretical properties of self-training algorithms.

In this line, Wei et al. (2021) suggest a new concept of *expansion* defined as the quantity of data dispersion in an example's neighbor, where the term *neighbor* refers to adding adversarial perturbations (Miyato et al., 2019) or augmentations (Sohn et al., 2020) to the example. The study establishes distributional guarantees of self-training when the label distribution meets such expansion properties and classes are suitably separated according to neighbors. The study generates finite sample bounds for Deep Neural Networks (DNNs) by combining generalization bounds with DNN generalization bounds. Experiments with a Generative Adversarial Network (GAN) are also used to verify the expansion assumption.

Frei et al. (2022) examine a self-training algorithm with linear models for the binary classification using gradient-based optimization of the cross-entropy loss after supervised learning with a small number of samples. The classifier is a mixture model with concentration and anti-concentration properties. The authors show that utilizing $O(d/\epsilon^2)$ unlabeled observations in the self learning algorithm, with $d$ the number of input variables, suffices to achieve the classification error of the Bayes-optimal classifier up to an $\epsilon$ error if the initial pseudo-labeling strategy has a classification error smaller than an absolute constant $C_{err}$. Furthermore, the authors demonstrate that a constant number of labeled examples is sufficient for optimal performance in a self-training algorithm by demonstrating that using only $O(d)$ labeled examples, the standard gradient descent algorithm can learn a pseudo-labeling strategy with a classification error no more than $C_{err}$.

Zhang et al. (2022) study the generalization ability of self-training in the case where the base classifier is a two-layer neural network with the second layer weights all fixed to one, and assuming that the ground truth is realizable, the labels are observed without noise, and the labeled and unlabeled instances are drawn from two isotropic Gaussian distributions. The authors show that, given some plausible assumptions about the initial point and the amount of unlabeled training examples, the algorithm converges to the ground truth with fewer observations than needed when no unlabeled data is provided. The reader can refer to Zhong et al. (2017) for a broader context. Zhang et al. (2022) extend their main result to a more general setting, where it is shown that the model still converges towards a given convex combination of the ground truth and the initial point, and is guaranteed to outperform the initial supervised model, without fixing any requirement on the number of labeled training examples.

Hadjadj et al. (2023) propose a first bound over the misclassification error of a self-training method which utilizes half-spaces as the base classifier in the case where class labels of examples are supposed to be corrupted by a Massart noise model. Under this assumption, it is shown that the use of unlabeled data in the proposed self-training algorithm does not degrade the performance of the first half-space trained over the labeled training data.

Sportisse et al. (2023) study the identifiability of self-training approaches. In addressing the bias in the conventional risk estimator, the proposed method, named Inverse Propensity Weighting, involves assigning weights to examples based on the inverse of their propensity scores-representing the probability of a class label being observed. The study

introduces two estimators for the missing data mechanism, one of which is derived through the maximization of the observed likelihood. Furthermore, a likelihood ratio test is suggested to evaluate the informativeness of the labels, determining whether they exhibit non-random missing patterns.

Some other works studied self-training from a theoretical perspective when a distribution shift takes place. Chen et al. (2020) proves that self-training can help to avoid spurious features, while Kumar et al. (2020) derived an upper-bound on the error of self-training in the case of gradual shifts.

## 3 Related and unrelated approaches

In semi-supervised learning, there are two main other areas of research that are related to self-training. The first, known as *transductive learning*, is based on the low density separation assumption and tends to give class labels for only the set of unlabeled training samples. The second method, referred to as *consistency learning*, uses classifier predictions over unlabeled data as a confidence indicator and constrains model outputs to be comparable for similar unlabeled examples without assigning pseudo-labels.

In this section, we also go a bit further, and introduce different context where self-training has been used and extended. First, we present *self-supervised learning*, which, despite its similar name with self-training, is an entirely separate technique that employs unlabeled data to train or pre-train a model. Finally, we introduce *reinforced self-training*, that merges elements of reinforcement learning with self-training principles by integrating a scoring function based on a learned reward model and employing offline reinforcement learning objectives for model fine-tuning.

### 3.1 Transductive learning

The goal of transductive learning, as previously stated, is to assign pseudolabels to samples from an unlabeled training set, $X_{\mathcal{U}}$. As this set is finite, the considered function class $\mathcal{F}$, for finding the transductive prediction function, is also finite. $\mathcal{F}$ can be defined using a nested structure according to the structural risk minimization principle, $\mathcal{F}_1 \subseteq \mathcal{F}_2 \subseteq \ldots \subseteq \mathcal{F}$ (Vapnik, 1998). Transductive techniques often employ the distribution of unsigned margins of unlabeled examples to guide the search for a prediction function, limiting it to following the low density separation assumption in order to find the best function class among the current ones.

Transductive approaches also assume that the function class's structure should reflect prior knowledge of the learning problem at hand, and that it should be built in such a way that the correct prediction of class labels of labeled and unlabeled training examples is contained in a function class $\mathcal{F}_j$ of small size with a high probability. In particular, Derbeko et al. (2004) show that with high probability the error on the unlabeled training set of a function from a class function $\mathcal{F}_j$ is bounded by its empirical loss on the labeled training set plus a complexity term that depends on the number of labeled examples $m$, the number of unlabeled examples $u$, and the size of the class function $\mathcal{F}_j$.

The Transductive Support Vector Machines (TSVM) (Joachims, 1999) developed for the binary case is based on this paradigm, and is looking for the optimal hyperplane in a feature space that separates the best labeled examples while avoiding high density areas. TSVM does this by building a structure on a function class $\mathcal{F}$ and sorting the outputs of unlabeled samples by their margins. The solutions to the associated optimization problem are the pseudo-labels of unlabeled examples for which the resulting hyperplane separates the examples of both labeled and unlabeled training sets with the largest margin.

Shi et al. (2018) extended this idea to the multi-class classification case with Neural Networks. Similar to TSVM, class labels of unlabeled examples are treated as variables, and the algorithm tries to determine their optimal values, along with the optimal NNs parameter set get by minimizing a cross-entropy loss estimated over both labeled and unlabeled training sets through an iterative training process. The authors employ the MinMax Feature regularization to constrain the neural network to learn features of same-class images to be close, and features of different classes to be separated by a preset margin, in order to overcome incorrect label estimations on outliers and noisy samples.

### 3.2 Consistency-based approaches

Early studies in this line, see for example Zhu et al. (2003) for binary classification, were proposed to learn a single classifier defined from a scoring function $f : \mathcal{X} \times \mathcal{Y} \to \mathbb{R}$ penalized for quick changes in its predictions. The

similarity matrix $\mathbf{W} = [W_{ij}]_{\substack{1 \leqslant i \leqslant u \\ 1 \leqslant j \leqslant u}}$, constructed over the unlabeled training data, is used to measure the similarity between instances. The penalization is mostly expressed as a regularization term in the learning objective. As an example, adapting the work of Zhu et al. (2003) to multi-class classification, the penalization term can be written as:

$$\Omega_{\mathbf{W}}(X_{\mathcal{U}}) = \sum_{i,j=1}^{u} W_{ij} \|f(\mathbf{x}_{m+i}, .) - f(\mathbf{x}_{m+j}, .)\|^2$$

where for a given example $\mathbf{x}$, $f(\mathbf{x}, .) = (f(\mathbf{x}, k))_{k \in \mathcal{Y}}$ is the vector class predictions of $f$. In terms of learning, $\Omega_{\mathbf{W}}$ can be seen as a regularization term, constraining the model to have the same predictions on similar unlabeled instances.

Other types of penalization have been studied in the literature. Maximov et al. (2018) suggested an approach that partitions partially labeled data and then uses labeled training samples to identify dense clusters having predominant classes with a fraction of non-predominant classes below a given threshold extending earlier results on supervised classification (Joshi et al., 2017). In this situation, the proposed penalization term measures the learner's inability to predict the predominant classes of the identified clusters which in turn constrains the supervised classifier to be consistent with the structure of the dense clusters.

In this line, Rangwani et al. (2022) consider non-decomposable metrics with consistency regularization by giving a cost-sensitive framework that consists of minimizing a cost-sensitive error on pseudo labels and consistency regularization. They demonstrate theoretically that they can build classifiers that can maximize the required non-decomposable measure more effectively than the original model used to produce pseudo-labels under comparable data distribution assumptions.

Without explicitly stating a penalization term, consistency learning was extended to cases with two classifiers. The Mean-Teacher approach (Tarvainen & Valpola, 2017) is perhaps one of the earliest popular techniques that have been proposed in this context. This method employs Neural Networks (NNs) as supervised classifiers, and it is based on the assumption that two close models with the same input should make the same prediction. One of the models is called the *teacher*, while the other is referred to as the *student*. These two NN models are structurally identical, and their weights are related in that the teacher's weights are an exponential moving average (Laine & Aila, 2017) of the student' weights. In this scenario, the student model is the only one that is trained over the labeled training set, and the consistency loss is computed between the teacher's probability distribution prediction and the student's one using the mean square error or the Kullback-Leibler divergence.

Other studies refined the Mean-Teacher approach with a data-augmentation technique by combining two images with random patches to improve prediction consistency (French et al., 2020; Xie et al., 2020a). More recently, Du et al. (2022) provide a two-stage method to reduce label propagation errors; where in the first phase, the gradients of the student loss are computed and utilized to update the teacher. In the second stage, the teacher assigns pseudo-labels which are then utilized to train the current student.

### 3.3 Self-supervised Learning

Although similar in names, self-training is a completely different approach than self-supervised learning which has demonstrated encouraging results and has become an active area of research (Ozbulak et al., 2023).

In self-supervised learning, a model acquires the ability to make predictions regarding various facets of its input data, all without the necessity of explicit labeled training data. Rather than depending on labeled data, self-supervised learning harnesses the inherent structure present in the input data and autonomously generates guidance to train the model. This procedure involves the formulation of a pretext task, also referred to as a proxy task, wherein the model is trained to make predictions concerning a specific relationship inherent in the data. For instance, in the domain of computer vision, a pretext task might involve rotating images within a predefined range of angles, followed by training a supervised model to predict these angles.

Once the model has undergone training on the pretext task, the knowledge it has gained in this process can be applied to downstream tasks that do require labeled data. Consequently, by learning from extensive amounts of unlabeled data, self-supervised learning empowers the acquisition of robust data representations, capitalizing on the abundant, freely available unlabeled data resources.

Common approaches in self-supervised learning include predicting missing parts of an image (Lee et al., 2021), predicting the order of shuffled image patches or their orientation (Shorten & Khoshgoftaar, 2019), reconstructing corrupted images (Fang et al., 2023), filling in missing words in a sentence (Donahue et al., 2020), or predicting future frames in a video sequence (Schiappa et al., 2022). These pretext tasks encourage the model to capture meaningful representations of the input data, which can then be used for various downstream tasks, such as image classification, object detection, or natural language processing.

### 3.4 Reinforced self-training

A recent innovative approach, called Reinforced self-training (ReST) has emerged, particularly notable for its application in conditional language modeling (Gulcehre et al., 2023; Singh et al., 2023). This approach operates through two distinct loops: the inner loop, called "Improve", which concentrates on refining the policy using a fixed dataset, and the outer loop, called "Grow", which involves expanding the dataset by sampling from the most recent policy.

In the domain of conditional language modeling, ReST follows a systematic sequence of steps. Initially, during the Grow phase, the language model policy, originally a supervised policy, generates multiple output predictions for each context, thereby enriching the training dataset. Subsequently, in the Improve stage, the expanded dataset undergoes ranking and filtering using a scoring function. The language model then undergoes fine-tuning on the refined dataset using an offline reinforcement learning objective, with the potential for repeating this process with an increasing filtering threshold. The resultant policy from this iterative process is subsequently employed in the following Grow phase.

ReST may find niche suitability in specific applications or scenarios where reinforcement learning principles enhance model performance through learned reward signals. In contrast, classical self-training techniques possess a broader applicability and have been employed across a wide spectrum of semi-supervised learning tasks without necessitating reinforcement learning frameworks.

## 4 Applications

In this section, we will concentrate on the most popular applications where self-training was employed, although this technique may be extended and used to a variety of additional machine learning tasks. The goal of our presentation here is not to be thorough, but rather to focus on the main features of self-training that were used in the literature among the selected applications.

### 4.1 Natural Language Processing

Co-training is perhaps one of the preliminary self-training techniques which was applied to web pages classification (Blum & Mitchell, 1998). In the paper, the content of a web page has been divided into two sets of words: those that appear on the page and those that appear in hyperlinks pointing to the page. The main hypothesis here is that each of the set of words contain sufficient information for the classification task and that there are enough labeled data to efficiently learn two supervised classifiers. Both theoretical and empirical studies of co-training show that if examples have two redundant but not entirely correlated views, then unlabeled data may be used to augment the original labeled training data to find more robust classifiers. However, the drawback of this strategy is that in general, text data is mono-view. For bag-of-word representation of texts, a solution was to split the set of words in two random sets, considered as two distinct views of a text (Nigam & Ghani, 2000), as mentioned in Section 2.4. However, this idea cannot be generalized to sequential models that could be used as base classifiers in co-training.

Other current self-training techniques in NLP are mostly built on the concept of co-training and employ two base classifiers that are learned over each other's predictions. In this line, Wu et al. proposed a Named Entity Recognition (NER) strategy that consists in automatically detecting and classifying named entities, with a first NER model trained on labeled training data serving as a teacher to predict the probability distribution of entity labels for each token in the unlabeled set. The pseudo-labeled data with such soft labels are then used to train a student NER model for the unlabeled set and the process of pseudo-labeling and training is repeated until convergence as in co-training. For the task of Relation Extraction (RE) which consists in obtaining a predefined semantic relation between two entities in a given sentence, Yu et al. (2022) proposed an approach which classifies the pseudo-labeled instances generated from

a teacher into confident, ambiguous and hard sets. In the training of the student model, the confident and ambiguous instances are subsequently interpreted as positive and set-negatives observations, respectively.

Lately, Meng et al. (2020) proposed an approach to leverage the power of language models that have been pre-trained on large corpora of text to generate pseudo-labels for unlabeled text data. The pseudo-labeled data along with a smaller set of labeled data are then used to train and fine-tune the text classifier, and the process of assigning pseudo-labels and retraining of the classifier is repeated until convergence. A challenge that arises when using a single base classifier in self-training for NLP tasks is to minimize the impact of label noise in the pseudo-labeling policy. To cope with this problem, Zadeh & Rasoul (2010) devised a bootstrapping technique for semantic role labeling that consists randomly selecting a subset of the most confident samples for pseudo-labeling. In the same vein, for the sentiment analysis task, Gupta et al. (2021) advocated selecting the top most confident samples for pseudo-labeling. However, as we shall see in the next section, the use of a fixed threshold in the pseudo-labeling policy may be suboptimal in general.

## 4.2 Computer Vision

As in NLP, the two variants of self-training with one or two classifiers, mainly referred to as student and teacher in the literature, are mainly considered for image classification. Most recent approaches use neural networks as base classifiers and rely on these models' ability to learn efficient representations of images, proposing various strategies to either improve the representation or reduce the effect of noise injection during the pseudo-labeling phase of self-training.

The most common strategy with student and teacher base classifiers is arguably the one proposed by Xie et al. (2020b), in which an EfficientNet model trained on labeled ImageNet images is used as a teacher to create pseudo labels on unlabeled ones. A larger EfficientNet is subsequently employed as a student model, being trained on a mix of labeled and pseudo-labeled images. This training involves altering the input images using various techniques like dropout, stochastic depth, and data augmentation. The objective is for the model to learn a representation of images that remains consistent despite these alterations. This procedure is repeated by reversing the roles of the student and the teacher. The input of the teacher model is not altered throughout the training process. The main motivation advanced is to ensure that the pseudo labels be as accurate as possible. Empirical evidence from various image collections demonstrates the effectiveness of this strategy.

Sohn et al. (2020) proposed a self-training approach called FixMatch that combines consistency regularization with a confidence-based mechanism to select high-confidence pseudo-labeled examples for training. The algorithm applies to the same image two different data augmentations procedures, called weak (flip-and-shift) and strong (more heavy distortions) augmentations. As in the previous case, these perturbations helps to increase diversity and improve the model's robustness on the unlabeled images. The authors introduce a consistency loss term that encourages the consistency between the model's hard output of the weakly-augmented version and the model's soft output of the strongly-augmented version of the same unlabeled image. They demonstrate that the model learns to provide more trustworthy and accurate results by minimizing the discrepancy between these predictions. In order to decrease the influence of possibly inaccurate pseudo-labels on the learning process, the loss is evaluated only on those unlabeled data from the batch that have the confidence higher than a fixed threshold.

This idea has then been adapted to various correlated tasks, including object detection, image segmentation (Cheng et al., 2023), remote sensing (Huang et al., 2023) and video anomaly detection (Lv et al., 2023), among others. Chen et al. (2022) proposed an improvement of FixMatch by introducing two novel features. First of all, they introduce a separate classification head that is used to assign pseudo-labels and trained using labeled data only in order to avoid possible label noise from wrong pseudo-labels. Secondly, they improve the feature learning by introducing an adversarial classification head whose goal is to approximate the worst possible error on unlabeled data. All these approaches employ a constant predefined threshold across all classes to choose unlabeled data for training, disregarding varying learning conditions and complexities among different classes.

To tackle this concern, Zhang et al. (2021) introduced a curriculum learning technique to utilize unlabeled data based on the model's learning progress. The essence of this strategy involves dynamically adapting thresholds for distinct classes during each time step, enabling the inclusion of insightful unlabeled data and their corresponding pseudo-labels. This approach has been successfully applied to many domains, including object detection (Li et al., 2022),

medical image classification (Peng et al., 2023), human action recognition (Wang et al., 2023) and facial expression identification (Shabbir & Rout, 2023).

## 4.3 Knowledge-driven applications

Through the incorporation of domain expertise, recent studies have developed more sophisticated self-training systems that reduce label noise in the pseudo-labeling phase across diverse applications. In the subsequent sections, we will consider advances made in this context in the domains of speech recognition, anomaly detection, genomics and proteomics.

### 4.3.1 Speech Recognition

Newly developed methods have introduced filtering mechanisms that are congruent with domain knowledge for end-to-end speech recognition. These mechanisms establish rules that assess pseudo-labels using criteria specific to the domain. For example by using filters to verify if certain phonetic patterns that are common in the domain, are present in the pseudo-labels (Gheini et al., 2023). Similar techniques incorporate phonetic information relevant to the domain to validate pseudo-labels. In these approaches, incorrectly labeled examples that violate phonetic constraints are discarded from training the model (Ling et al., 2022).

Other approaches integrate domain-specific language models in the the pseudo-label generation process in order to ensure that the generated labels adhere to the linguistic nuances and terminologies of the domain. In this line, Kahn et al. (2020) introduced a self-training approach, with one base classifier combined with a language model for pseudo-labeling. Their approach involves implementing tailored filtering methods designed to address common errors arising from sequence-to-sequence models, alongside an inventive ensemble technique for enhancing the breadth of pseudo-label variations. Building upon this idea, Xu et al. (2021) showcased that the synergy between self-training and unsupervised pre-training using wav2vec 2.0 (Baevski et al., 2020) offers mutual benefits across diverse scenarios involving labeled and unlabeled data.

As in image classification, alternative methods for speech recognition apply data-augmentation techniques, tailored to the unique aspects of the domain, to enhance the robustness of the model's predictions and consequently the quality of pseudo-labels. In this sense, Bartelds et al. (2023) employed a text-to-speech system to generate audio training data from text-only sources.

### 4.3.2 Anomaly Detection

Leveraging domain knowledge to mitigate label noise in pseudo-labels within self-training approaches has also been considered in anomaly detection. In this case, the understanding of the anomaly patterns and characteristics specific to the domain are incorporated in the model. In this regard, Li et al. (2012) identified common anomaly types, their features and potential sources of noise and Qu et al. (2023) performed time domain analysis. Also, Feng et al. (2021) created features that capture domain-specific information for video anomaly detection. It was demonstrated that these features highlight the crucial elements for anomaly detection in videos, leading to their utilization for enhancing the pseudo-labeling phase within self-training. Alternate strategies focus on simulating anomalies within the unlabeled dataset using domain knowledge. This aids the model in learning from a broad spectrum of anomalies, mitigating the potential of becoming overly specialized in a particular anomaly type (Qiu, 2023).

### 4.3.3 Genomics and proteomics

Furthermore, datasets in the field of genomics and proteomics encompass a variety of characteristics including gene expression levels, epigenetic markers, and genetic variants. These characteristics have been shown to increase the effectiveness of features used in self-training approaches, together with the selection of important features and their physiologically coherent transformation.

Brubaker et al. (2019) incorporated biological context into feature engineering that integrate unsupervised modeling of datasets relating to human disease with the supervised component that concentrated on training with mouse data. In this context, Ravinder (2021) amalgamated expression data from three distinct humanized mouse models that were subjected to live attenuated yellow fever vaccine challenges in self-training with different base classifiers. The results

of this study show that self-training coupled with NRG-HIS/Fluc mice exhibited the most favorable outcomes across the tested human cohorts.

Additionally, El-Manzalawy et al. (2016) employed self-training in conjunction with bioinformatic tools in silico to anticipate secreted and protective proteins. This was done to eliminate pseudo-label errors from the identified P. falciparum SEPs obtained through proteomics experiments and to anticipate new SEPs within the P. falciparum proteome.

Huang et al. (2021) applied domain-specific quality control steps to clean and pre-process the data. This included filtering out low-quality samples, normalizing data to account for technical biases, and addressing batch effects that can introduce noise. By doing so, they ensured that the unlabeled data that is feed into the self-training pipeline is as accurate as possible. Chan et al. (2017) utilized reference databases and annotation resources related to genomics. These resources provide information about genes, functional elements, pathways, and biological processes. Incorporating this information into the pseudo-labeling process has been shown to lead to more accurate predictions by aligning them with known biological knowledge. Yu et al. (2023) applied network analysis techniques to identify interactions between genes and proteins. The authors demonstrated that Pathway enrichment analysis can help identify genes that are functionally related and likely to be co-regulated. This information has been shown to guide the self-training process to produce more coherent and biologically plausible pseudo-labels.

**General observations**    The key observations made in these applications reveal that, in pseudo-labeling, employing fixed thresholds often yields suboptimal outcomes, underscoring the importance of dynamic thresholding for optimal results. Furthermore, enhancing pseudo-label noise improves both generalization and class differentiation. In Appendix A, we will show the impact of dynamic thresholding on pseudo-labeling across general benchmarks proposed in Feofanov et al. (2019) and examine the noise considerations in two image classification collections studied in Chen et al. (2022).

## 5    Conclusion and Perspectives

In this survey, we provided an overview of self-training approaches for semi-supervised learning that have received increasing attention in recent years.

First, we discussed the various strategies for selecting unlabeled samples for pseudo-labeling that have been proposed. We emphasized the significance of considering margin distributions across unlabeled data as a pivotal factor in the development of these strategies. Next, we provided an overview of the diverse variants of self-training explored in the literature, along with relevant approaches. Furthermore, we examined recent theoretical advancements in this research domain and outlined the principal characteristics of self-training employed in several widely recognized applications. Lastly, we explored the impact of fundamental aspects of self-training on a range of benchmark datasets.

While the self-training approach is currently in widespread use, there are extensive opportunities for future research. Presently, the majority of studies have concentrated on perturbation-based deep learning, particularly in the domains of visual, text, and audio applications. However, there exist numerous other domains, such as industrial time-series or medical data, where the application of self-training could prove highly beneficial.

Recent research emphasizes the potential of exploring self-training methods from a theoretical standpoint, particularly in addressing the challenge of training a final classifier on data with noisy labels Hadjadj et al. (2023). It has also been demonstrated that accurately estimating the confidence of pseudo-labels is crucial for effective self-training (Odonnat et al., 2024). Therefore, theoretically establishing the correlation between performance and the level of uncertainty in pseudo-labeling could be a valuable direction for future research, especially in analyzing self-training within the context of learning problems affected by domain shifts.

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

| Data set | # of labeled examples $m$ | # of unlabeled examples $u$ | Dimension $d$ | # of classes $K$ |
|---|---|---|---|---|
| Vowel | 99 | 891 | 10 | 11 |
| Protein | 129 | 951 | 77 | 8 |
| PageBlocks | 1094 | 4379 | 10 | 5 |
| Isolet | 389 | 7408 | 617 | 26 |
| HAR | 102 | 10197 | 561 | 6 |
| Pendigits | 109 | 10883 | 16 | 10 |
| Letter | 400 | 19600 | 16 | 26 |
| Fashion | 175 | 69825 | 784 | 10 |
| MNIST | 175 | 69825 | 784 | 10 |

Table 2: Characteristics of data sets used in our experiments, $d$ and $K$ correspond to respectively the dimension of the input space and the number of classes.

# A Empirical Study

Within this section, we will evaluate the effectiveness and performance of the self-training algorithm. This assessment will be based on various key features presented in the preceding sections, and it will be conducted across multiple benchmark scenarios. Our primary focus will be on scenarios characterized by severely limited labeled training data, where the utilization of complex baseline classifiers like deep learning models is unfeasible. Additionally, we will address the prevalent scenario where there are sufficient labeled training data, enabling the development of an initial supervised complex model.

**The impact of threshold selection.** We first study the effect of selecting automatically the threshold for pseudo-labeling on 9 publicly available data sets proposed for semi-supervised learning (Dua & Graff, 2017). The characteristics of these datasets are presented in Table 2. It is worth noting that certain datasets contain only a limited number of labeled training examples, comprising just a few hundred instances and accounting for less than 1% of the total training examples. This condition underscores the suitability of employing complex base classifiers.

In the experimentation, Random Forest was employed instead using the scikit-learn implementation (Pedregosa et al., 2011) with 200 trees of maximum depth while leaving other parameters at their default values. The primary objective was to assess and contrast the classifier's performance in two scenarios: the supervised scenario (denoted by RF) and the self-training scenario where pseudo-labeling is automatically conducted following the approach introduced by Feofanov et al. (2019)[1] (denoted by PL\*). Additionally, we investigated the impact of setting the pseudo-labeling threshold at predefined values from the set $\theta \in \{0.5, 0.7, 0.9\}$ (denoted by $PL_\theta$).

The automatic pseudo-labeling strategy selects the threshold which minimizes the bound of the error of the Random Forest classifier over the unlabeled training samples.

Results are resumed in Table 3. Experiments are repeated 20 times by choosing randomly the labeled training examples, and $\downarrow$ indicates that performance is statistically worse than the best result, shown in bold, according to the Wilcoxon rank-sum test.

These results suggest that the effectiveness of self-training heavily relies on the method used to determine the pseudo-labeling threshold. When the threshold is automatically determined, self-training (i.e. PL\* ) can perform competitively, indicating that this approach has the potential to improve results compared to the supervised RF.

However, when a fixed threshold is applied, self-training tends to yield inferior results compared to the supervised learning approach. This suggests that an arbitrarily chosen threshold might not effectively capture the underlying patterns in the data for the pseudo-labeling process, leading to suboptimal performance.

Moreover, when the threshold is too low as for $\theta \in \{0.5, 0.7\}$, pseudo-labeling is likely to produce label noise and degrade the performance of self-training with respect to the supervised RF classifier in all cases. When the threshold it is too high (i.e. $\theta = 0.9$), self-training becomes competitive compared to RF on Isolet and MNIST, but the quantity of pseudo-labeled unlabeled examples seems not to be sufficient to learn efficiently.

---

[1] https://github.com/vfeofanov/trans-bounds-maj-vote

| Data set | RF | PL $_{\theta=0.5}$ | PL $_{\theta=0.7}$ | PL $_{\theta=0.9}$ | PL$^\star$ |
|---|---|---|---|---|---|
| `Vowel` | $\mathbf{.586} \pm .028$ | $.489^{\downarrow} \pm .016$ | $.531^{\downarrow} \pm .034$ | $.576^{\downarrow} \pm .028$ | $\mathbf{.586} \pm .026$ |
| `Protein` | $.764^{\downarrow} \pm .032$ | $.653^{\downarrow} \pm .024$ | $.687^{\downarrow} \pm .036$ | $.724^{\downarrow} \pm .018$ | $\mathbf{.781} \pm .034$ |
| `PageBlocks` | $.965 \pm .003$ | $.931^{\downarrow} \pm .003$ | $.964 \pm .004$ | $.965 \pm .002$ | $\mathbf{.966} \pm .002$ |
| `Isolet` | $.854^{\downarrow} \pm .016$ | $.648^{\downarrow} \pm .018$ | $.7^{\downarrow} \pm .04$ | $.861^{\downarrow} \pm .08$ | $\mathbf{.875} \pm .014$ |
| `HAR` | $.851 \pm .024$ | $.76^{\downarrow} \pm .04$ | $.81^{\downarrow} \pm .041$ | $.823^{\downarrow} \pm .035$ | $\mathbf{.854} \pm .026$ |
| `Pendigits` | $.863^{\downarrow} \pm .022$ | $.825^{\downarrow} \pm .022$ | $.839^{\downarrow} \pm .036$ | $.845^{\downarrow} \pm .024$ | $\mathbf{.884} \pm .022$ |
| `Letter` | $.711 \pm .011$ | $.062^{\downarrow} \pm .011$ | $.651^{\downarrow} \pm .015$ | $.673^{\downarrow} \pm .015$ | $\mathbf{.717} \pm .013$ |
| `Fashion` | $.718 \pm .022$ | $.625^{\downarrow} \pm .014$ | $.64^{\downarrow} \pm .04$ | $.68^{\downarrow} \pm .014$ | $\mathbf{.723} \pm .023$ |
| `MNIST` | $.798^{\downarrow} \pm .015$ | $.665^{\downarrow} \pm .012$ | $.705^{\downarrow} \pm .055$ | $.823^{\downarrow} \pm .045$ | $\mathbf{.857} \pm .013$ |

Table 3: Classification performance using the accuracy score on 9 publicly available data set. Best results are shown in bold and the sign $\downarrow$ shows if the performance is statistically worse than the best result on the level 0.01 of significance.

In summary, the findings emphasize the importance of a dynamic and adaptive threshold selection mechanism when implementing self-training.

**Noise Account.** We now consider the case where the initial labeled training set allows to train deep neural networks and examine the effects of taking into account noise in the pseudo-labeling process along with the dynamic selection of the threshold on CIFAR-10 and CIFAR-100 (Krizhevsky & Hinton, 2009). Both datasets contain 32x32 pixel RGB images belonging to respectively 10 and 100 classes; 50000 examples are used for training and 10000 samples for test.

We consider the debiased self-training approach (`DST`) (Chen et al., 2022) to address the presence of noise in pseudo-labeling, in conjunction with the FlexMatch method (Zhang et al., 2021) for the dynamic threshold determination in pseudo-labeling. As outlined in Section 2.3, `DST` involves training a dedicated head on pseudo-labeled examples, allowing the model to implicitly capture and account for noise inherent in the pseudo-labels.

For FlexMatch, we followed the same experimental protocol than Zhang et al. (2021). In this case, Wide ResNet (`WRN`) (Zagoruyko & Komodakis, 2016) was used as the base classifier in self-training. Parameter learning was accomplished using stochastic gradient descent (SGD) with a momentum coefficient of 0.9. The initial rate was set to $\eta_0 = 0.03$ with a cosine learning rate decay schedule as $\eta = \eta_0 \cos(7\pi t/16T)$, where $t$ denotes the current training step and $T$ is the total training step set at $2^{20}$. Additionally, exponential moving averaging with a momentum of 0.999 was implemented and the batch size for labeled data was fixed to 64. For `DST`, we used the code made available by the authors[2].

We compared FlexMatch with and without the `DST` approach denoted respectively by `FM` and `FM+DST`. We also compared self-training with `WRN` trained in fully supervised manner. Each experiment was repeated 5 times by changing the seed at each time. Figure 2 presents the average accuracy of different models on the test set for the same number of initial labeled training samples per class within the set $\{4, 10, 20, 50\}$ for both datasets. In both datasets, considering label noise within pseudo-labels leads to improved performance, with the improvement being more pronounced in the case of CIFAR-100.

In CIFAR-100, classes are structured into 20 superclasses, each comprising 5 related classes, addressing noise in this more complex task aids in class differentiation and enhances the model's ability to generalize. It is worth noting that with a greater number of initial labeled training examples, the gap between the `FM` and `FM+DST` approaches narrows, as the model becomes more proficient with the increased labeled data and makes fewer errors in pseudo-labeling.

---

[2] https://github.com/thuml/Debiased-Self-Training

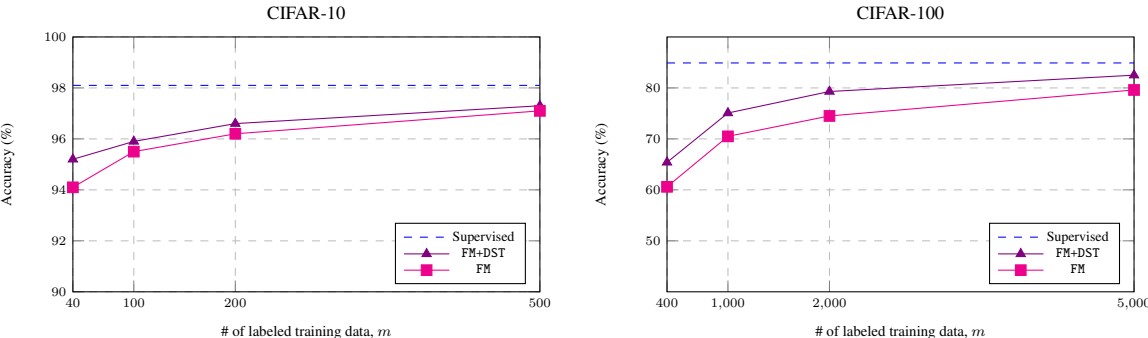

Figure 2: Comparisons in terms of Accuracy on CIFAR-10 and CIFAR-100 for a varying number of labeled training data. "Supervised" refers to the fully supervised learning ($m = 50000$, $u = 0$).

