# OpenReview forum: "Self-Training: A Survey"
_TMLR — Rejected by TMLR_

### Review · Reviewer_LXk5 · 2024-02-11

**Summary Of Contributions:**

This paper presents a comprehensive review of self-training methods for both binary and multi-class classification tasks, along with their variants and two related approaches. Additionally, the authors focus on prevalent applications leveraging self-training techniques and suggest avenues for future research. The authors assert that this paper constitutes the first extensive and exhaustive survey on self-training methodologies.

**Audience:**

Yes

**Claims And Evidence:**

Yes

**Requested Changes:**

Refer to the Weaknesses

**Strengths And Weaknesses:**

Strengths:
1. This paper serves as a comprehensive review, featuring a well-structured framework encompassing sections dedicated to framework introduction, methodology, application scenarios, challenges, conclusion, and perspectives.
2. This paper is the first overview and summary of the self-training.

Weaknesses:
1. This paper mainly discusses 14 algorithms within the self-training. However, the classification of these algorithms lacks clarity. It is suggested that the author explicitly categorize the algorithms to enhance readers' understanding of the research landscape in the self-training domain.
2. The structure of the paper is not sufficiently rigorous. Moreover, the author fails to emphasize the logical relationships between relevant methodologies, leading to suboptimal coherence in the paper.
3. Redundancy issues are apparent in this paper. For example, the content of the section introduction does not align closely with the subsequent presentation of methodologies. It is recommended that the author streamline the content and dedicate specialized sections to elaborate on the foundational knowledge of semi-supervised learning and self-training.
4. The references exhibit formatting issues, with some papers lacking page numbers. Additionally, the author overlooks the inclusion of several crucial references pertinent to self-training. It is advised that the author rectify the formatting of the references and supplement the paper with the relevant omitted literature.

   [a] Self-training with noisy student improves imagenet classification. CVPR 2020

   [b] Multi-task self-training for learning general representations. ICCV 2021

---

> ### Author Response · Authors · 2024-02-19
> **Response about the structure**
>
> We thank the reviewer for the advice to structure better the paper.
>
> We have reorganized Table 1 to emphasize the classification of the algorithms, and structure the paper into
> 1/ Introduction
> 2/ Self-training
> 3/ Related and unrelated approaches
> 4/ Applications
> 5/ Conclusion and perspectives
> Which is clearer and less redundant.
>
> We check the formatting issues in the bibliography, and add the two references suggested by the reviewer.

---

### Review · Reviewer_MFmj · 2024-02-12

**Summary Of Contributions:**

This paper focuses on the self-training framework,  which is to learn a classifier iteratively by assigning pseudo-labels to the set of unlabeled training samples with a margin greater than a certain threshold. This paper reviews the self-training methods for binary and multi-class classification as well as
their variants and two related approaches. With the empirical studies on CIFAR10/CIFAR100, this paper provides an examination of the impact of factors of the self-training framework. In addition, the related application and future research are discussed.

**Audience:**

Yes

**Claims And Evidence:**

No

**Requested Changes:**

Please consider the above "other comments/questions" part to revise the draft.

**Strengths And Weaknesses:**

Strengths:
- This paper studies the self-training problem which is relevant in many applications.
- This paper provides an investigation of the related work of self-training methods from different perspectives and discusses the application as well as the impact of self-training features.

Weaknesses:
- The major claims or findings of the current survey are not clear, and readers can hardly get the main information that the authors want to present.
- The current presentation can be improved by re-organizing the subsections with an overview to guide the readers to a better understanding of some commonalities and differences.
- The empirical studies and references can be better added for a survey paper.

Other comments/questions:
- Please specify "this subject" at the end of the abstract.
- It could be better to explicitly summarize the results or information of the presentation or examination of the self-training methods or the impact of significant self-training features in the abstract and introduction. Otherwise, the readers can not clearly catch up with the current version's unique discoveries or major claims.
- Please add more references (citations) in the introduction part to support each sentence with claims.
- Is there any rigorous definition for "smoothness"? It could be better to provide some illustrative figures to present the central hypothesis.
- Is there any relationship among the three main semi-supervised learning families? The writing of this part can be further improved by considering summarizing it as a table.
- The current notation of a part of the unlabeled data is not appropriate, can we use $\bar{u}$?
- For each part from Sec.3.1 to Sec.3.3, it could be better to draw some important message after simply explaining each work.
- Except for the benchmark CIFAR10/CIFAR100, are there any other results on different datasets or large-scale datasets (like ImageNet)? The current empirical studies are not very convincing in supporting the discovery
- The current presentation has a large room to improve by better structuring and summarization.

---

> ### Author Response · Authors · 2024-02-19
> **Response about experiments**
>
> We appreciate your comment concerning the experimental section and your suggestions for clarification.
>
> The primary aim of including experiments in the  paper was to gather the two key features closely related to self-training presented in the literature, which are the selection of the threshold and taking into account the noise. However, we realize that the contribution of this section might not have been explicitly articulated, as also pointed out by reviewer ``nd8o''.
>
> To address this concern, we propose to push the experiments that can be found on other papers in the appendix, accompanied by a clear statement outlining the key features they represent in the paper.

---

> ### Author Response · Authors · 2024-02-19
> **Response about clarification**
>
> We have re-written added content to the abstract and the introduction by examining the impact of significant self-training features. Furthermore, we added more citations in the introduction and added a figure illustrating the three main hypotheses in semi-supervised learning.
>
> We have changed the name of the pseudo-labeled set to $\bar{U}$ and clarified the relationship among the three main semi-supervised learning families.

---

> ### Author Response · Authors · 2024-02-19
> **Response about structure**
>
> We have reorganized Table 1 to emphasize the classification of the algorithms, and structure the paper into 1/ Introduction 2/ Self-training 3/ Related and unrelated approaches 4/ Applications 5/ Conclusion and perspectives Which is clearer and less redundant.

---

### Review · Reviewer_nd8o · 2024-02-17

**Summary Of Contributions:**

- A survey paper focused on self-training within semi-supervised learning. This provides an opportunity for the paper to go deeper, e.g., discuss the theoretical studies of self-training.
- Up-to-date: includes many papers from 2022 and 2023, which is what other survey papers do not cover.

**Audience:**

Yes

**Broader Impact Concerns:**

I do not have any ethical concerns.

**Claims And Evidence:**

Yes

**Requested Changes:**

I would appreciate it if the authors can read the 2 points I raised in the weaknesses.

Some other minor comments are:

- Period at the end of page 1 is missing.
- Page 2: filed --> field
- Period missing at the end of the 2nd last paragraph of page 4
- Table 1: Only the MeanTeacher paper has the method name, while all other papers have author names. Can we make them consistent?
- Period missing, last sentence, 3rd paragraph of Section 7

**Strengths And Weaknesses:**

Strengths

Although there are already many survey papers on semi-supervised learning, this survey paper focuses on self-training, which makes it a unique survey paper.

The survey includes discussions about self-training from various aspects, e.g., theoretical studies of self-training and applications of self-training to different fields such as NLP and computer vision.

Weaknesses

The title is "Self-training: A survey", but the paper focuses on self-training from the perspective of semi-supervised learning. In my opinion, a more precise title may be "Self-Training in Semi-supervised Learning: A survey". If the paper can expand the discussions about self-training beyond semi-supervised learning, I think the current title will become a great fit. For example, the paper discusses domain adaptation in just one paragraph in the final section, but this should be discussed in depth if the title does not limit the survey to semi-supervised learning. Self-training may also include topics such as knowledge distillation where teacher/student are identical (Furlanello et al., 2018) and is a hot topic in language models (Singh et al. 2023)

Furlanello et al.: https://arxiv.org/abs/1805.04770

Singh et al.: https://arxiv.org/abs/2312.06585

The contribution of Section 6 is not clear. In the first set of experiments (the impact of threshold selection), the experiments seem quite similar to the experiments in Feofanov et al. (2019). Feofanov et al. (2019) already discusses how sensitive the threshold used for pseudo-labeling can be for the final performance. In the second set of experiments: this also seems similar to the experiments in Chen et al., 2022, where they combine FlexMatch and DST. It would be great if the authors can clarify the contributions.

---

> ### Author Response · Authors · 2024-02-19
> **Response about the title**
>
> We appreciate your recommendation for a more precise title, with respect to the new Reinforced self-training approach that merges elements of reinforcement learning with self-training principles by integrating a scoring function based on a learned reward model and employing offline reinforcement learning objectives for model fine-tuning. These methodological distinctions distinguish reinforced self-training from classical self-training, which typically relies on iterative pseudo-labeling and model retraining without reinforcement learning components.
>
> We prefer to retain the current title, as it is the most apt reflection of the overarching theme and scope of our work. Indeed, classical self-training has long been a staple technique in the realm of semi-supervised learning, preceding reinforced self-training and finding extensive application across various domains such as NLP or computer vision.
>
> Although the first focus of our study is on self-training methods in the framework of semi-supervised learning, as you rightly pointed out, we acknowledge the importance of presenting related subjects like domain adaptation and reinforced self-training in Section 4, that we now call "Related and unrelated approaches". To address the need for clarity in distinguishing between different self-training paradigms, we have incorporated a paragraph in Section 4 elucidating these distinctions and a subsection for each application.
>
> We hope that this clarification adequately addresses your concerns regarding the title while affirming our commitment to maintaining coherence and relevance in the presentation of our research. Thank you once again for your insightful comments, which have undoubtedly contributed to the refinement of our work.

---

> ### Author Response · Authors · 2024-02-19
> **Response for the experiments**
>
> We appreciate your comment concerning the experimental section and your suggestions for clarification.
>
> The primary aim of including experiments in the  paper was to gather the two key features closely related to self-training presented in the literature, which are the selection of the threshold and taking into account the noise. However, we realize that the contribution of this section might not have been explicitly articulated, as also pointed out by reviewer  ``MFmj''.
>
> To address this concern, we propose to push the experiments that can be found on other papers in the appendix, accompanied by a clear statement outlining the key features they represent in the paper.

---

> ### Author Response · Authors · 2024-02-19
> **General response**
>
> We thank the reviewer for the interesting comment about our paper.
> We have corrected other (minor) concerns of the reviewer in the revised version of the paper.

---

### Author Response · Authors · 2024-02-19
**general comment**

We thank all the reviewers and the action editor for their valuable comments.
We have responded to each reviewer concerns, and will update a pdf of the revised paper by Wednesday, the 21st.

---

### Decision · Action_Editor_q1rG · 2024-03-19

**Recommendation:** Reject

**Comment:**

The submission surveyed the self-training approach to semi-supervised learning. After the rebuttal and revision, one reviewer voted for acceptance (nd8o) and the other two voted for rejection. Their post-rebuttal comments can be found below.

> (Reviewer *nd8o*) Thank you for updating the paper. The paper pushed the experiments to the Appendix, but the contributions of the experiments remain unclear. It would be better to explicitly explain the differences between the experiments in this paper and the experiments in Feofanov et al. (2019), Chen et al., (2022).

> (Reviewer *LXk5*) After reading the revision and the reviews of other reviewers, I have reached a decision leaning toward rejection. This decision is grounded in several aspects. Firstly, the authors have not adequately addressed the weaknesses. For example, the logical coherence among pertinent methodologies remains deficient, and there exist missing page numbers or inaccurate publication years in the references. Secondly, the revision does not include experimental results of the self-training algorithms, which is a crucial aspect of the comprehensiveness of a survey paper. Lastly, I agree with Reviewer nd8o's comments regarding the inadequacy of the paper's title, which does not aptly reflect its contents.

> (Reviewer *MFmj*) Summary: This paper focuses on the self-training framework, which is to learn a classifier iteratively by assigning pseudo-labels to the set of unlabeled training samples with a margin greater than a certain threshold. This paper reviews the self-training methods for binary and multi-class classification as well as their variants and two related approaches. With the empirical studies on CIFAR10/CIFAR100, this paper provides an examination of the impact of factors of the self-training framework. In addition, the related application and future research are discussed.
>
> After carefully reading the comments from the other reviewers and the response from the authors, I'm leaning reject for the current version of this work for the following reasons:
>
> (1) The structure of this paper remains incoherent, making the readers lost in the claims from different perspectives although the analysis is comprehensive. For example, the newly revised Section 1.4 can still not emphasize the logical relationship between relevant methodologies. The empirical study seems not closely related to the previous method description.
>
> (2) The author's response fails to address the reviewer's comments clearly. There is no point-to-point response to all the reviewers' requested changes, making the second round review hard to distinguish the differences from the first version and also hard to understand the exact response corresponding to the previous weaknesses.
>
> (3) Some requested challenges are not fully addressed, making some claims remain unclear. For example, the authors didn't explicitly explain the "smoothness" and there is limited summarization in each section to convey the major information of the survey results. For the experiments, the empirical study is not sufficient and not insightful, it is also pointed (by reviewer nd8o) as similar to previous related works.
>
> Overall, the reviewer thinks the current presentation still has room to improve by considering all the unsolved issues.

I don't think we could accept the current version of this submission to be published on TMLR. In particular, I strongly agree with nd8o and suggest you carefully address the comments from nd8o about the title issue and the experiment issue (as a survey paper, you cannot assume that a reader looking at your title knows that self-training is for semi-supervised learning and different from self-supervised learning/training). The first post-rebuttal comment from MFmj may also be very important for improving the quality of this manuscript.

**Audience:**

Yes, this is a survey paper about a hot topic in semi-supervised learning.

**Claims And Evidence:**

No, this is a survey paper, and there are not new scientific claims to be supported by evidence.

If we consider claims here to be some specific organization/classification and scientific opinions about self-training theory, algorithms, and applications, there is not enough evidence why the ones made by the authors in this survey is better or more reasonable than existing ones (at least the reviewers and I didn't see the evidence) given that the research topic of self-training can be dated back to 1970s.

**Resubmission Of Major Revision:**

The authors may consider submitting a major revision at a later time.